# Maritime Transport Network in Korea: Spatial-Temporal Density and Path Planning

**Jeong-Seok Lee, Tae-Hoon Kim and Yong-Gil Park ***

Marine Bigdata & AI Center, Korea Institute of Ocean Science & Technology, Taejong-ro, Yeong-do, Busan 49112, Republic of Korea; jslee90@kiost.ac.kr (J.-S.L.); thkim00@kiost.ac.kr (T.-H.K.)
* Correspondence: ygpark32@kiost.ac.kr

**Abstract:** The increase in maritime traffic and vessel size has strengthened the need for economical and safe maritime transportation networks. Currently, ship path planning is based on past experience and shortest route usage. However, the increasing complexity of the marine environment and the development of autonomous ships require automatic shortest path generation based on maritime traffic networks. This paper proposes an efficient shortest path planning method using Dijkstra's algorithm based on a maritime traffic network dataset created by extracting maritime traffic routes through a spatial-temporal density analysis of large-scale AIS data and Delaunay triangulation. Additionally, the depth information of all digital charts in Korea was set as a safety contour to support safe path planning. The proposed network-based shortest path planning method was compared with the path planning and sailing distance of a training ship, and compliance with maritime laws was verified. The results demonstrate the practicality and safety of the proposed method, which can enable the establishment of a safe and efficient maritime transportation network along with the development of autonomous ships.

**Keywords:** path planning; spatial-temporal density; maritime transportation; network analysis; Delaunay triangulation

## 1. Introduction

Owing to the worldwide increase in maritime traffic and vessel size, the importance and connectivity of maritime transportation routes have become increasingly significant [1]. This implies that the marine transportation environment is becoming increasingly diverse, leading to an increasing demand for the analysis of maritime transportation routes [2]. Maritime transportation routes generally refer to areas where ships operate. Several such routes can be interconnected to form a maritime transportation network [3]. Maritime transportation networks provide various routes through the connection of ports for maritime logistics [4]. Lee and Cho stated that automatic identification system (AIS) data are essential for various methods of analyzing maritime transportation [5]. The International Maritime Organization (IMO) mandates the installation of AIS data on ships for safe navigation and environmental protection [6]. AIS data include information such as ship type, ship size, date and time, ship location, and ship specifications, and record ship tracks from the past [7]. By analyzing the AIS tracks of ships engaged in international navigation, optimal strategies for sailing from the origin to the destination can be planned based on the ship type and size [8]. The optimal route is economical and safe for ship operators, and most ships exhibit similar sailing patterns [9]. According to Akdağ et al. and Tan et al., the development of unmanned navigation technology leads to the active development of maritime autonomous surface ships (MASSs) and unmanned surface vehicles (USVs) [10,11]. Currently, ship operations are conducted through navigation planning and the actions of navigators and captains based on their experience and judgment. However, because of the decreasing number of seafarers, new navigation planning methods based on

computer algorithms using AIS data, which have been accumulated over the years, have become essential, particularly owing to the increasing complexity of maritime traffic and the development of MASS technology [12–14].

In this study, a novel method for shortest path navigation planning is proposed through AIS data analysis to address the need for more efficient and safer navigation planning for ships. A spatial-temporal density analysis was performed using the AIS data accumulated over several years, and the navigational areas of the ships were represented as polygons. The AIS data underwent transformation from points to lines, and the analysis involved calculating the temporal values occupied within each grid, resulting in the spatial-temporal density analysis. Delaunay triangulation analysis was performed based on these polygons to divide maritime traffic areas into separate spatial units, using the traffic separation scheme (TSS) method to differentiate two-way traffic. The Delaunay triangulation results involved a process of filtering out areas, excluding those with high spatial-temporal density values. The resulting spatial units constituted a maritime traffic network dataset of about 800, which was supplemented with depth information from a digital chart to prevent grounding. The digital chart was based on the S-57 hydrographic data generated for navigation purposes. Dijkstra's algorithm was used to analyze the navigation time and distance based on the proposed method to plan a safe and efficient route from the departure point to the destination. The proposed method was compared with the path planning of a training ship in terms of the navigation time, distance, and compliance with TSS regulations. The AIS data, along with the spatial-temporal density analysis and the maritime traffic network dataset generated using space partitioning algorithms, were analyzed to better represent the shortest path planning compared to the results of the reference training ship. The proposed maritime network dataset and depth information are expected to enhance the safety of ship navigation and can be used for the future navigation planning of MASSs and USVs.

## 2. Related Study

In this section, we classify the various analyses of maritime traffic into studies on maritime traffic route generation, maritime traffic network proposals, and ship path planning to represent the shortest distance. Additionally, we describe the main algorithms used in the analysis, including the method of spatial partitioning.

### 2.1. Studies on Maritime Traffic Routes

Lee et al. aimed to create a national maritime traffic map for South Korea's coastal waters [5]. The study focused on cargo ships, tanker ships, passenger ships, and fishing vessels with lengths of 60 m or more, using AIS data for four weeks (28 days). A line density analysis was performed by connecting ship tracks and extracting the top 50% of the data using quantile partitioning to derive maritime traffic routes. Similarly, Kim et al. conducted a study on maritime traffic routes in the same area with four months (120 days) of AIS data and applied the same ship types and sizes [15]. They proposed a spatial-temporal density analysis method based on the occupied time of ship tracks within the analysis area. This method addressed the limitations of line density analysis and enabled the analysis of large amounts of maritime traffic data. This method utilized the approach provided by the European Marine Observation and Data Network (EMODnet) for ship density, which is publicly available [16]. Maritime traffic routes represent a way to depict high-density maritime traffic areas by using polygons to represent the areas, rendering them suitable for various applications.

### 2.2. Studies on Maritime Traffic Network

Pallotta et al. explained the effectiveness of using AIS data to infer spatial and temporal information for port and ocean platforms as the usage of AIS data increases [17]. They proposed a Traffic Route Extraction and Anomaly Detection (TREAD) method, which reviewed the utilization of this knowledge for the characteristics of maritime traffic, the



extraction of traffic routes, and other related analyses based on the distribution analysis of maritime traffic. Fernandez et al. extracted maritime traffic networks using unsupervised methods on ship trajectory data, decomposing them into significant routes by detecting waypoints, and constructing maritime traffic networks by connecting ports and waypoints via graph-based connections [18]. Wang et al. extracted dense maritime traffic areas via the kernel density estimation method, and used image processing to detect their edges [19]. They proposed a polygon-based approach to represent areas, and generated centerlines connecting nodes and segments via Delaunay triangulation. Yan et al. transformed the rich positional information of ship trajectories into semantic objects called "ship trip semantic objects (STSO)", which represent the objects as a "stop-waypoint-stop" model [20]. They utilized graph theory to integrate the nodes and edges for maritime traffic network construction. Filipiak et al. utilized ship traffic volume derived from AIS data to construct maritime traffic networks [21]. Their method comprises three steps: maneuvering point detection, waypoint discovery, and edge construction, utilizing the k-d B-tree and Quadtree algorithms for spatial partitioning to extract waypoints. Graph-based maritime traffic networks were constructed using a genetic algorithm based on the extracted waypoints. The construction of a maritime traffic network that creates a path from a ship's origin to its destination is fundamental.

### 2.3. Studies on Path Planning

Shah et al. studied path planning for USVs operating in complex environments [22]. They utilized the A* algorithm at the visibility graph nodes, and employed quadtree spatial partitioning to efficiently calculate the nodes of the visibility graph. Various distance-based cost-to-go functions were proposed. Lee et al. also utilized quadtree to perform visibility graph-based path planning for USVs in Korean coastal waters [23]. The nodes generated through the quadtree were used to create a quadtree-based graph, and Dijkstra's algorithm was applied to extract the shortest path. Lee et al. used the depth data to detect the shortest maritime traffic routes for AIS-based ships [24]. Depth is the most important parameter for preventing maritime accidents when planning ship routes. In that study, a grid-based navigational area was extracted, and Dijkstra's, A*, and improved A* algorithms were used for the shortest path calculation. Prior to using the shortest path algorithm, they proposed a suitable algorithm for merchant ships that considered factors such as depth, restricted areas, and designated routes.

### 2.4. Research Improvement Plan

Merchant ships in international shipping generally use comfortable and safe routes to prevent accidents. Because navigation is impossible in certain areas, depending on the size of the ship, directly searching for the shortest route can reduce the usability of actual ship operations. In other words, the areas in which small vessels navigate are different from those in which large vessels navigate. This study presents the shortest route that can be applied to ships of a size capable of engaging in international navigation and suggests the following improvement measures.

In contrast to previous studies, this study utilized a large-scale AIS density analysis as basic data to represent maritime traffic areas as polygons. The high-density areas obtained from the density analysis provide evidence that ships have used them safely and economically. Therefore, we aim to produce maritime traffic routes based on the studies of Lee et al. and Kim et al. to represent the areas where ships primarily operate [5,15]. The polygons representing the navigation area of a ship can be used to create routes for ships by utilizing the Delaunay triangulation algorithm.

We differentiate our study by separating the routes, such that merchant ships can use them in two ways. Most maritime traffic networks represent single-line connections. However, major accidents are likely if ships operate in different directions along the same route. Therefore, when constructing maritime traffic networks, we assumed that ships could use these networks in a two-way manner.

Merchant ships engaged in international shipping must comply with mandatory navigation regulations. We built a network based on the premise of safe ship navigation by enforcing TSSs and using depth data.

## 3. Materials and Methods

The AIS data comprising spatial information are essentially a maritime traffic dataset including various attribute information [25]. Spatial information can be used as a basis for various analyses such as maritime traffic pattern recognition, maritime traffic prediction, and maritime traffic networks using geographic information systems (GIS) [26]. In this study, ArcGIS Pro 3.0.3 was used to generate maritime traffic routes, perform delay triangulation, build network datasets, and perform network analysis. To differentiate it from previous studies, AIS data were used as the basis for maritime traffic density analysis. Furthermore, a network dataset was developed based on ArcGIS to store the routes of various ships in the marine space, and a network analyst was used [27]. Additionally, the digital chart's shapefile files were combined with network analysis for practical ship use. Considering safety, adherence to navigation regulations, and accident prevention, we propose a shortest path maritime traffic network.

### 3.1. Study Overview

The entire Korean coastal area was selected to construct the maritime traffic network dataset, as shown in Figure 1. Previous research has focused on relatively small areas near ports or shallow waters for shortest path analysis. In contrast, this study aims to construct a national-scale maritime traffic network for ships to connect ports. Korea is located in Northeast Asia, and its geography depends strongly on the maritime industry, with 99.7% of its water volume used for shipping [28]. Therefore, the construction of a maritime traffic network is urgently required to handle the smooth flow of maritime traffic, and the importance of safe and efficient shortest path planning is increasing. This study emphasizes the need to construct a comprehensive maritime traffic network that covers all areas.

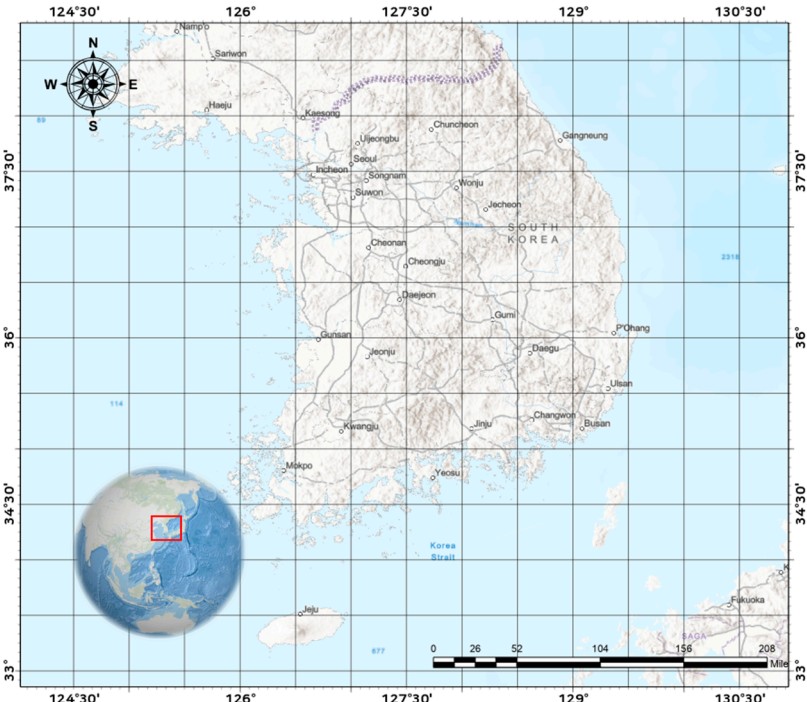

**Figure 1.** Detailed location and the analysis area.

The process for deriving the shortest path through a GIS-based network analysis is shown in Figure 2. Appropriate preprocessing is required because the initial AIS data are not directly usable. The AIS data, which are composed of point-based spatial information, are transformed into line-based information based on the ship's trajectories over time. Subsequently, a spatial-temporal density analysis is performed to identify the dense routes that are commonly used by ships. The resulting density can be represented as a polygon of the shipping area, and the Delaunay triangulation method is used to partition the area into triangles that are suitable for building a maritime traffic network. Subsequently, three equidistant points were inserted into each line segment to regulate the traffic flow of the ships. These points are subsequently connected in sequence to create a maritime traffic network, which is stored as a network dataset. This dataset, including safety contours and navigation regulation information from digital charts, enables the derivation of the shortest path through network analysis.

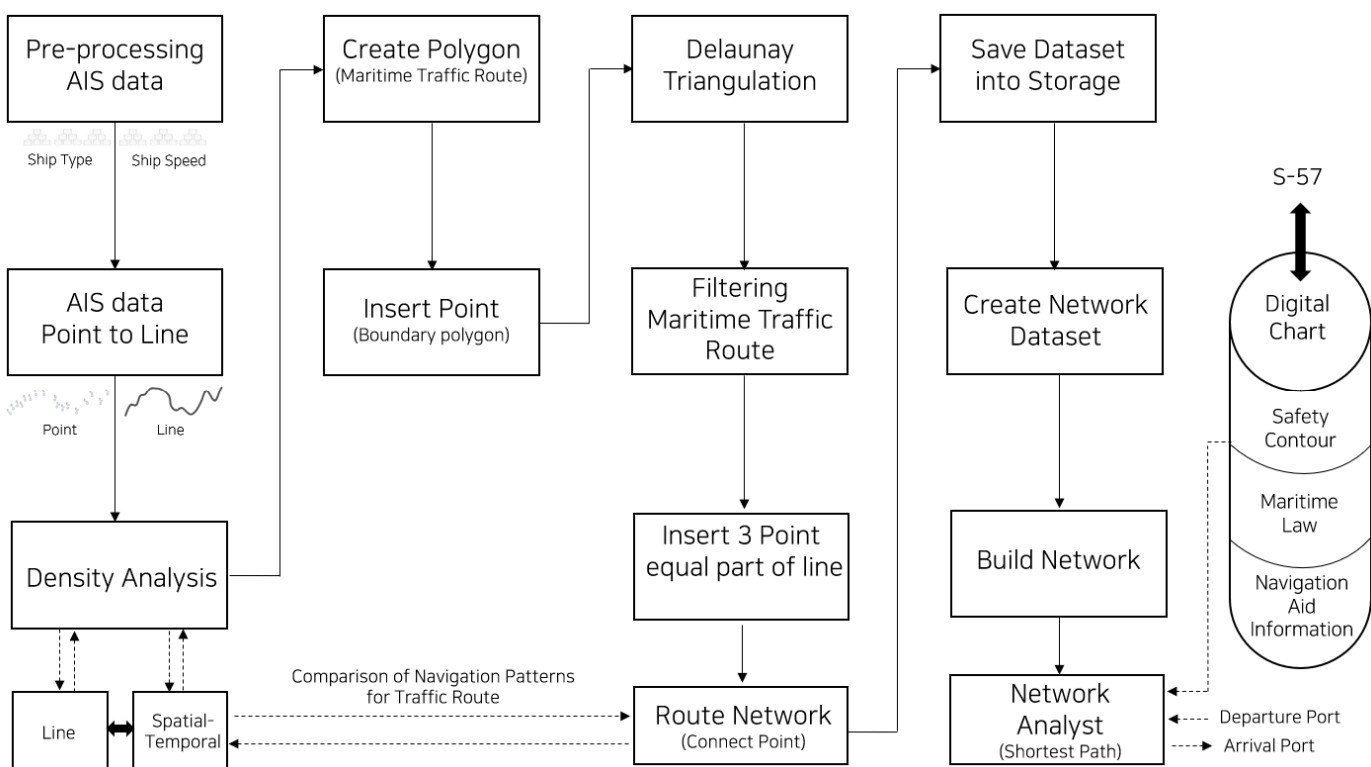

**Figure 2.** Flowchart of shortest path planning in maritime transport using network analysis.

### 3.2. Overview of AIS Data and Density Analysis

AIS data are recorded in real time, and the AIS is equipment installed to support the safe navigation of ships by providing information regarding their operations and specifications [29]. AIS data afford several advantages for maritime transportation analysis, including real-time data acquisition, high accuracy, and detailed ship information such as ship type, speed, and location, rendering them a valuable tool for improving navigation safety and optimizing shipping operations. In addition, AIS data can provide insight into maritime traffic patterns and trends, which can aid the development of effective maritime policies and regulations. The collected AIS data are divided into static and dynamic information. Static information includes the maritime mobile service identity (MMSI), name, type, IMO number, call sign, length, draft, and gross tons (GTs), whereas dynamic information includes the MMSI, date, latitude, longitude, speed over ground (SOG), course over ground (COG), and heading [30]. Preprocessing was performed by combining the static and dynamic information based on the MMSI data for analysis. In this study, the AIS data collected in Korea during the four seasons of 2018 were used for analysis, and

data from March 1 to 7 (1 week), June 1 to 7 (1 week), September 1 to 7 (1 week), and December 1 to 7 (1 week) were used. According to Tsuji (1996), maritime traffic survey data should consider a minimum of 6–7 days of traffic volume, accounting for weekly fluctuations [31]. Therefore, this study performed a maritime traffic density analysis for 28 days, using 7 days of data for each season. Various types of ships use the sea. However, in this study, cargo ships, tanker ships, passenger ships, and towing ships were targeted to construct a network dataset for major shipping routes. Figure 3 shows the results of the spatial-temporal density analysis of AIS trajectories and traffic zones within the density area in polygon form. The spatial-temporal density analysis method used in Figure 3a follows the analyses used by Kim et al. and EMODnet, which are widely used scientific methods for ship density providing public services in Europe and Korea [15,16].

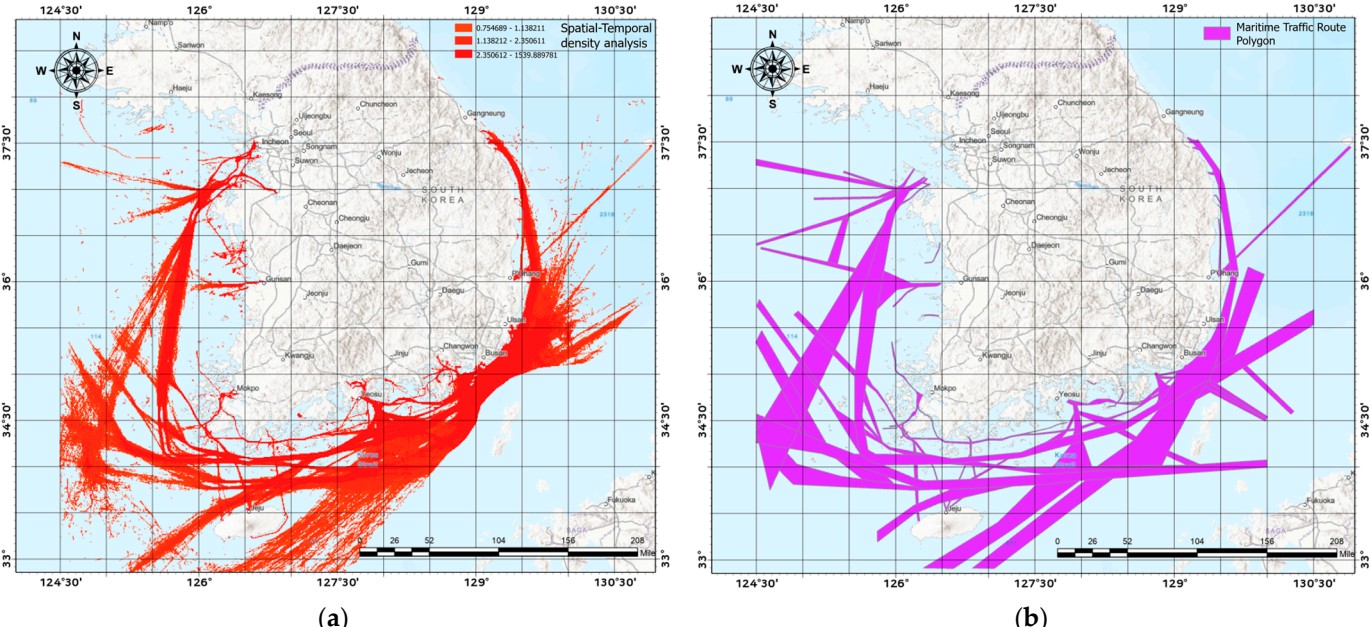

**Figure 3.** Visualization of AIS data analysis: (**a**) results of spatial-temporal density analysis targeting AIS; and (**b**) polygon results of maritime traffic area including spatial-temporal density analysis results.

The density was calculated using Equation (1). When a line between two consecutive positions on a ship intersects two or more cells, the length of the line segment crossing these cells can be calculated. To calculate the time for which a ship occupied each cell, the length of the line segment was divided by the total length of the line and subsequently multiplied by the total time of the line. The density of each individual cell is subsequently determined by calculating the time value of each line segment and adding all the time values associated with the cell.

$$D_i = \sum_{j=1}^{n} \frac{S_j}{L_j} \times T_j \tag{1}$$

where $D_i$ is the ship density (h) of cell $i$, $L_j$ is the total length (km) of the line $j$, $S_j$ is the partial length (km) of the line $j$ that intersects with cell $i$, and $T_j$ is the time (h) spent by each ship in each cell during the entire period (e.g., one month) and the number of lines related to the cell. Therefore, the density is the time (hours) each ship spends in a given cell over the entire period. The polygon representing the area where ships navigate in Figure 3b is included in the spatial-temporal density analysis result. This area can change and vary depending on changes in the maritime traffic volume and traffic flow trends according to the season. As the polygon is not a fixed element but can vary depending on the situation,

it can be used in other analysis areas. In this study, the polygon, as shown in Figure 3b, was used to perform Delaunay triangulation.

### 3.3. Method for Performing Delaunay Triangulation Based on Density Analysis

Based on the density analysis, extracting the area where ships conventionally operate as polygons is feasible. This area is not only the flow area for the safe navigation of ships but also serves as a basis for further analysis. Delaunay triangulation constructs a polygon in the form of triangles based on points inserted at regular intervals on the edge of a maritime traffic area. Delaunay triangulation is a method for constructing and dividing triangles in a manner that excludes points other than those used to create the triangle. One of the most important features of Delaunay triangulation is that "the circumcircle of each triangle does not contain any points other than the three vertices of the triangle". This feature is useful for identifying the closest point, rendering it useful for data clustering, density analysis, and road network design [32]. Figure 4a shows the initial results of Delaunay triangulation on the generated polygon, and Figure 4b shows the filtered triangles generated outside the polygon area.

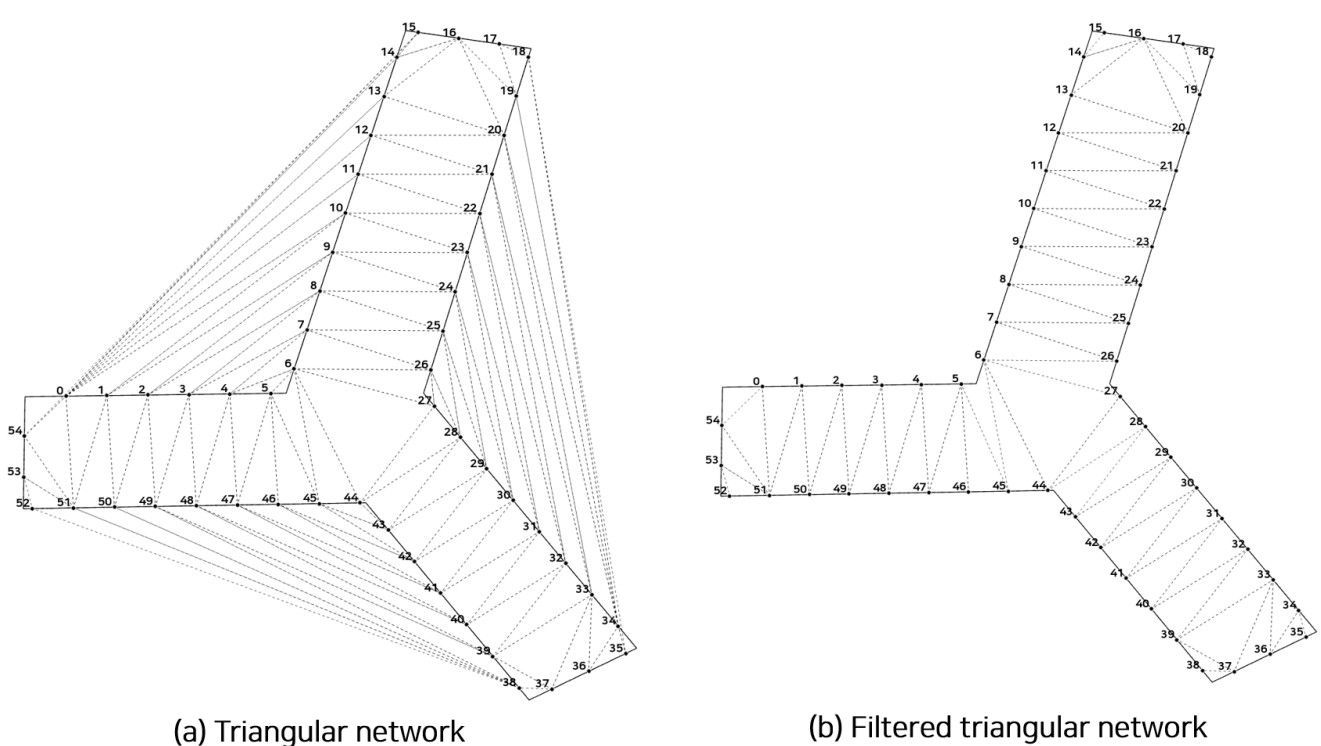

(a) Triangular network      (b) Filtered triangular network

**Figure 4.** Process of generating Delaunay triangulation: (**a**) Delaunay triangulation before filtering; and (**b**) Delaunay triangulation after filtering.

Maritime traffic in ocean spaces exhibits complex traffic flows depending on the types and sizes of ships. Shipping lanes for cargo ships have been established as a safe and efficient means of navigation, rendering the analysis of cargo ship density in Korean coastal waters important. The results of the density analysis and Delaunay triangulation for cargo ships in Korean coastal waters are shown in Figure 5.

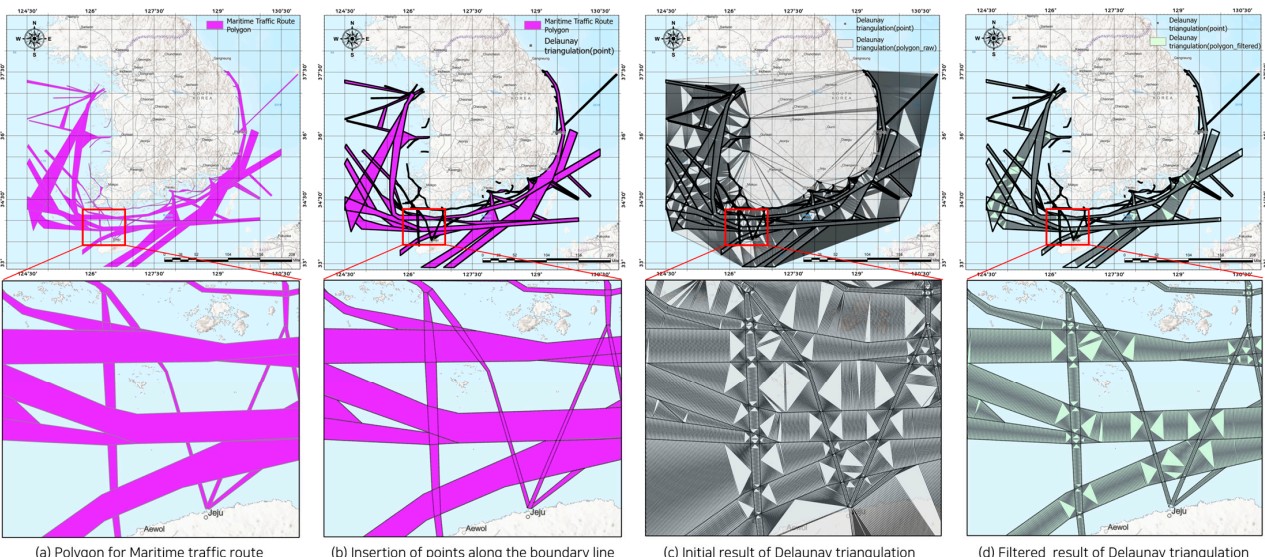

(a) Polygon for Maritime traffic route  (b) Insertion of points along the boundary line  (c) Initial result of Delaunay triangulation  (d) Filtered result of Delaunay triangulation

**Figure 5.** Results of the density-based Delaunay triangulation performed on the coastal waters of Korea: (**a**) spatial-temporal density-based polygon representing the maritime traffic flow, (**b**) insertion of points at regular intervals along the edges of the polygon, (**c**) initial results of Delaunay triangulation before filtering, and (**d**) results after filtering.

Figure 5a shows the initial state of the maritime traffic route and illustrates the results of the spatial-temporal density analysis extracted as polygons. In Figure 5b, point insertion along the boundary lines of the extracted polygons is performed at regular intervals. The points are inserted at a consistent interval of 1 km, equivalent to the grid size used in the density analysis. Figure 5c shows the results of Delaunay triangulation, encompassing areas beyond those extracted as maritime traffic routes. Subsequently, Figure 5d presents the filtering of the triangulation in land areas, retaining only the portions corresponding to maritime traffic routes. Maritime traffic flows differ depending on the types and sizes of ships. The analysis targeted the shipping routes for merchant ships that operate safely and efficiently, rendering them important elements of maritime traffic.

### 3.4. Network Analysis Method Using Digital Chart Data

The digital chart refers to all hydrographic information related to ship navigation, such as coastlines, contour lines, water depths, navigational aids, hazards, and shipping routes, produced according to the International Hydrographic Organization (IHO) standard specification S-57. It provides path planning, route monitoring, and navigation-related information for safe ship navigation [33]. The inclusion of digital chart information in network datasets is essential for safe ship navigation. In this study, we aim to create a new maritime traffic network dataset that includes depth and navigational information that can be utilized for actual ship operations. Figure 6 shows the detailed information on the S-57-based digital chart used for the maritime traffic network dataset.

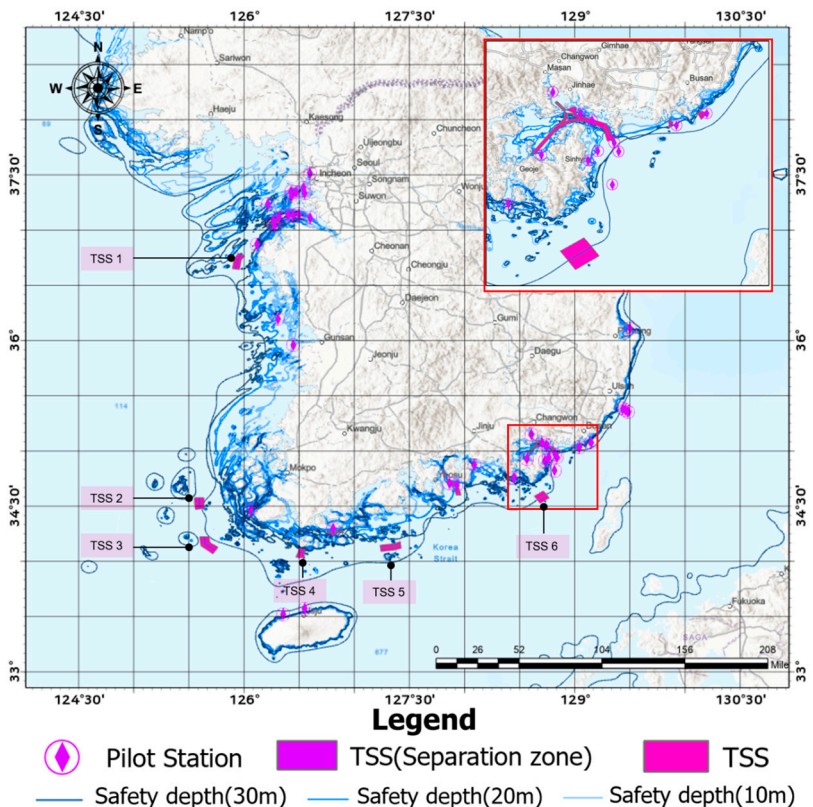

**Figure 6.** Detailed description of S-57-based digital charts for the maritime traffic network (pilot area, TSS, depth).

In general, a pilot boards a vessel to enter or leave a port. Similarly, for ships to leave the port, pilots board and safely unberth the vessel [34]. Thus, ships can navigate freely from the point of departure to their destination outside the port area, utilizing the most economical and safe routes. Therefore, the Pilot Station (P/S) can be used as information for departures and destinations. TSS is designed to prevent collisions between ships at sea and has rules for navigation according to the designated flow [35]. The navigation rules of the TSS are specified in Part B, Section I, Rule 10, Traffic Separation Schemes, as shown in Figure 7.

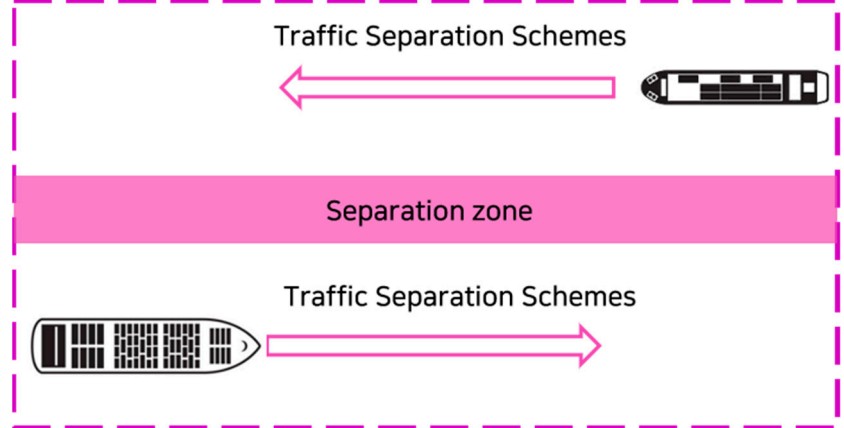

**Figure 7.** Detailed description of Traffic Separation Schemes (TSSs).

To ensure safe navigation for ships, TSSs are designated in wide areas of the sea beyond routes to ports of call. TSS was established to promote safe navigation in areas with heavy maritime traffic. The direction of the TSS should be considered when building a maritime

transportation network to ensure the safe navigation of ships. Furthermore, the size of a ship significantly affects the factors considered in its navigation. Depth information is a critical factor that must be considered, particularly for larger ships. A larger ship requires a wider water area for maneuvering, rendering depth a sensitive factor. Therefore, depth data are crucial for preventing ship grounding accidents. In this study, depth information was applied to the network dataset at intervals of 10 m, 20 m, and 30 m. To secure a safe depth for ship navigation, depth information is set as a line obstacle in the shortest path detection algorithm.

*3.5. Construction Method of Network Dataset Based on Delaunay Triangulation*

Previous studies using Delaunay triangulation to construct maritime traffic networks connected these networks based on centerlines. A route based on centerlines is suitable for representing a single traffic flow but not for expressing various navigation patterns. The polygon inside the maritime traffic route includes not only ships sailing in a certain direction but also upstream, downstream, eastbound, westbound, and irregular navigation patterns. To induce regular maritime traffic flow, it is regulated to connect in the same direction as the TSS. Because TSS is a method for regulating bidirectional maritime traffic flow, constructing a bidirectional maritime traffic network dataset using Delaunay triangulation is necessary. Figure 8 illustrates the process of creating a bidirectional maritime traffic network that regulates maritime traffic in both directions.

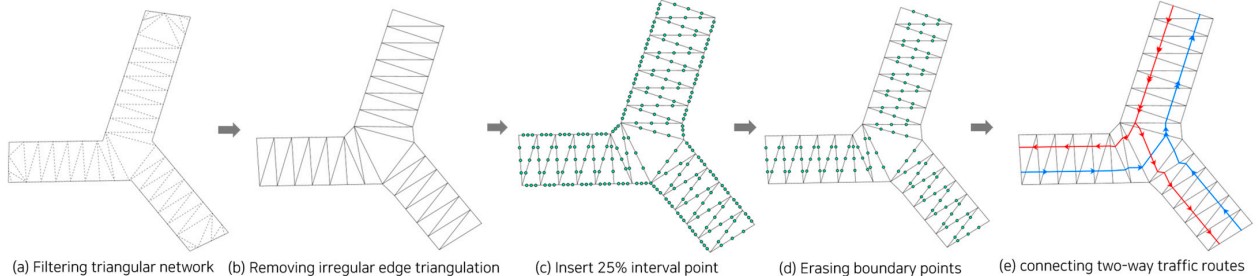

(a) Filtering triangular network  (b) Removing irregular edge triangulation  (c) Insert 25% interval point  (d) Erasing boundary points  (e) connecting two-way traffic routes

**Figure 8.** Construction process of two-way maritime traffic routes using Delaunay triangulation: (**a**) Initial results of Delaunay triangulation filtering, (**b**) Removal of irregular triangulation at the edges, (**c**) Insertion of points at 25% intervals in the triangulation results, (**d**) Removal of points along the boundary line, (**e**) Connection of two-way traffic routes.

Figure 8a shows the process of converting the initial delay triangulation, which is formed as a polygon, into line-based spatial information. Figure 8b shows the process of deleting the irregular edge triangulation, which enables forming an accurate maritime traffic route. Figure 8c inserts points at a distance of 25% for each line, resulting in three points being generated for each line, which represent the points that indicate the flow of maritime traffic. Figure 8d shows the process of deleting the points formed on the boundary by uniformly utilizing the difference in spatial information. Afterwards, Figure 8e forms three points only for the lines that exist inside, where the left point connects the south and west bounds and the right point connects the north and east bounds. The points formed in the middle are used as centerlines for the maritime traffic network route in this study. However, they can be created separately according to the separation provided by the TSS. Thus, maritime traffic flow is divided into two-way forms, preserving the regular flow of ships and focusing on accident prevention through the installation of separate buoys. The maritime traffic network dataset is constructed based on the created two-way route. The process of building by reorganizing network connections and attribute information is subsequently performed on the input network dataset. This implies rebuilding the included routes, such that they are available for use in the network, and the work speed can vary significantly depending on the data size during this process.

## 4. Results

### 4.1. Extraction Results of Network Based on Delaunay Triangulation

Various types of digital chart information have been considered when constructing maritime traffic networks. Maritime traffic includes areas with one-way navigation and areas where navigation is enforced to be two-way. Additionally, stations are available where pilots are required to board and disembark to enter and exit the ports. Lastly, safe water depths for the target vessels must be considered, as the permissible depths vary depending on the size of the ships. The ultimate goal of a maritime traffic network is to safely and economically transport large volumes of maritime cargo using ships, while considering various factors [36]. The maritime traffic network that can be established in South Korea's coastal waters is listed in Table 1. After constructing the maritime traffic network dataset, 485 datasets were generated based on six major trade ports located in the west (Incheon, Pyongtaek, Daesan, Boryeong, Gunsan, and Mokpo), with a total route length of 323,351 km and an average route length of 666.7 km.

**Table 1.** Results of maritime traffic network dataset.

| Area | Number of Ports (Names) | Number of Route Networks | Route Total Distance (km) | Route Average Distance (km) |
|---|---|---|---|---|
| Western | 6 (Incheon, Pyongtaek, Daesan, Boryeong, Gunsan, and Mokpo) | 485 | 323,351 | 666.7 |
| Southern | 5 (Jeju, Wando, Yeosu, Busan Newport, and Busan port) | 153 | 56,684 | 370.5 |
| Eastern | 3 (Ulsan, Pohang, and Donghae) | 100 | 61,053 | 610.5 |
| Outside | 18 | 62 | 28,662 | 462.3 |
| Total | 32 | 800 | 469,750 | 527.5 |

In the south, five major trade ports (Jeju, Wando, Yeosu, Busan Newport, and Busan port) generated 153 datasets with a total route length of 56,684 km and an average route length of 370.5 km. The three major trade ports in the east (Ulsan, Pohang, and Donghae) generated 100 datasets with a total route length of 61,053 km and an average route length of 610.5 km. In addition to the network dataset of routes departing from major trade ports, data approaching trade ports from offshore were added as "outside". A total of 62 "outside" datasets were generated, with a total route length of 28,662 km and an average length of 462.3 km. Here, the term "outside" refers to routes approaching coastal areas from the open sea. Therefore, 32 starting points were selected in the coastal waters of South Korea, and 800 network datasets were constructed. Finally, density analysis was performed using the AIS data. The results of the maritime traffic network dataset based on Delaunay triangulation are shown in Figure 9.

The construction of a maritime traffic network must consider the actual flow of ship traffic, comply with regulations such as those designated in the TSS, and ensure a safe water depth to prevent grounding accidents. The established network dataset was subsequently used as the basis for performing shortest path analysis.

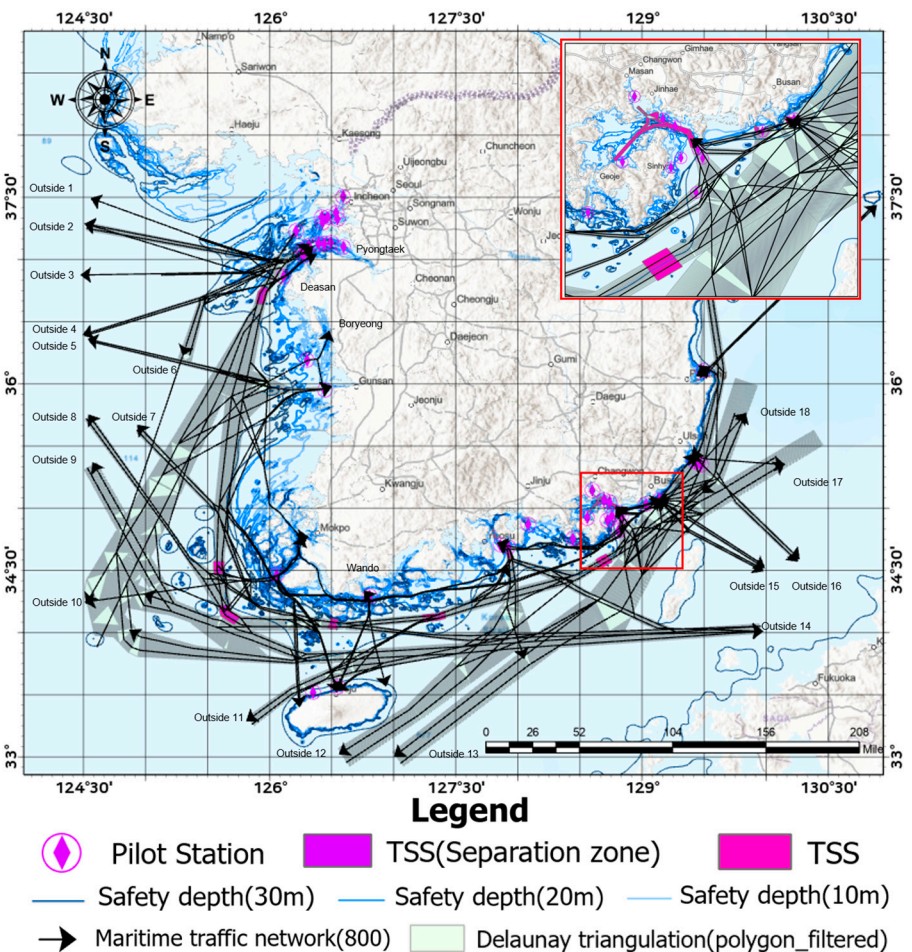

**Figure 9.** Result of the construction of the maritime traffic network dataset (32 starting points, 800 network datasets).

### 4.2. Shortest Path Network Analysis for Voyage Planning

For ships to navigate from the departure point to the destination, considering factors such as the economically shortest route, safe water depth, and compliance with designated routes is necessary. Dijkstra was the first to devise a method for generating the shortest route based on graph theory, and Silveira et al. and Wang et al. constructed the shortest route that ships can use using Dijkstra's algorithm [37–39]. In this study, Dijkstra's algorithm was used to calculate the shortest route, and the safety depth data were set as line obstacles in the network. This implies that by simultaneously inputting line obstacle values to maintain a safe water depth during ship navigation, the network dataset provides the shortest route among the available routes. Using Dijkstra's algorithm provided by ArcGIS Network Analyst, the safe area is set by considering points (marine facilities), lines (depth), and polygons (military training), and a safety check can be performed for the designated routes.

### 4.3. Comparison of Shortest Path Planning Algorithms

To compare the actual path planning of the training ship (T/S HANBADA) with the shortest path planning, the ship's specifications are listed in Table 2, and its photograph is presented in Figure 10.

**Table 2.** Specifications of T/S HANBADA.

| Category | T/S HANBADA |
| --- | --- |
| Date built | 8 December 2005 |
| Length (overall) | 117.20 (m) |
| Beam | 17.80 (m) |
| Draught | 8.15 (m) |
| Gross tonnage | 6686.0 (tons) |
| Service speed | 17.5 (knots) |
| Main engine | Diesel 8130 (HP) |

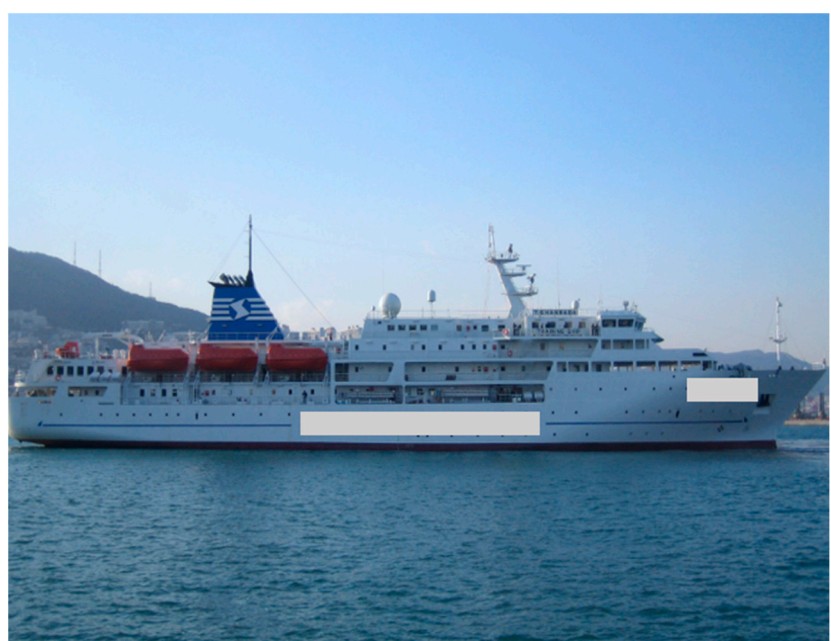

**Figure 10.** Photograph of T/S HANBADA.

The training ship T/S HANBADA is currently in operation as a vessel for student onboard education. The ship's length is 117.20 m, the beam is 17.80 m, and the draught is 8.15 m, which is an important factor for designating the safety depth as the depth to which a vessel is immersed in water. Therefore, a 10 m depth boundary line was entered as a line obstacle in the maritime traffic network dataset. To compare the shortest path and compliance with designated routes provided by the network dataset, the path planning used in the actual operation of the training ship (Case A) from the P/S of Busan port to the P/S of Incheon port and the path planning (Case B) from the P/S of Incheon port to the P/S of Busan port were compared. Figure 11 shows the training ship's route, and Figure 12 shows the network dataset-based path planning connecting the P/S of the Busan and Incheon ports, two major trading ports in Korea.

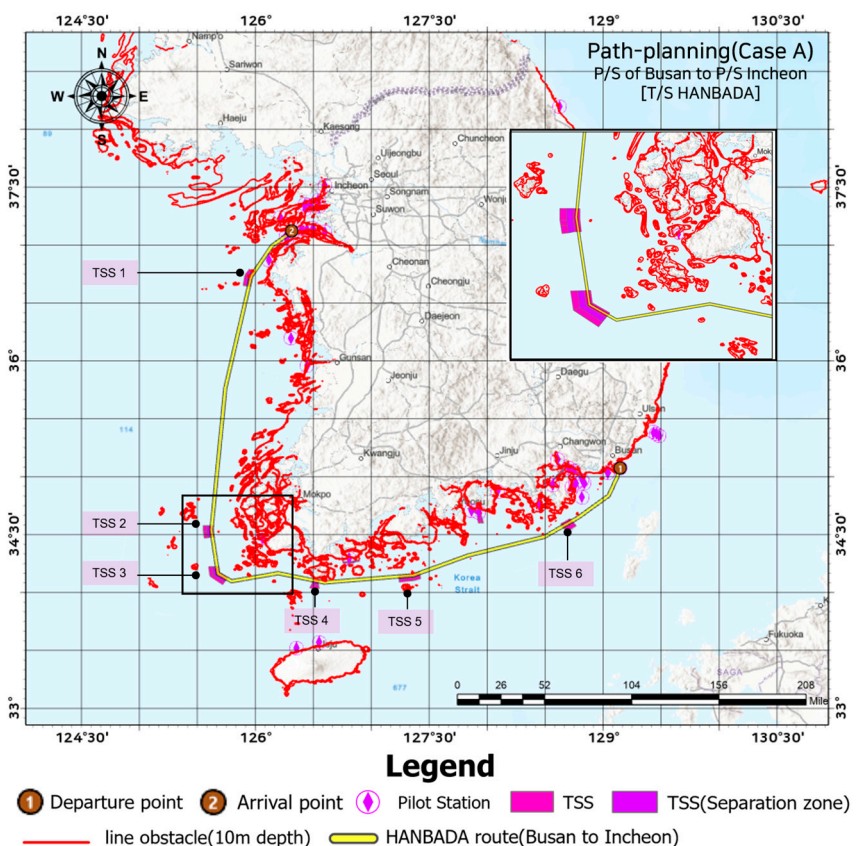

**Figure 11.** Path planning of the training ship (Case A) from Busan to Incheon.

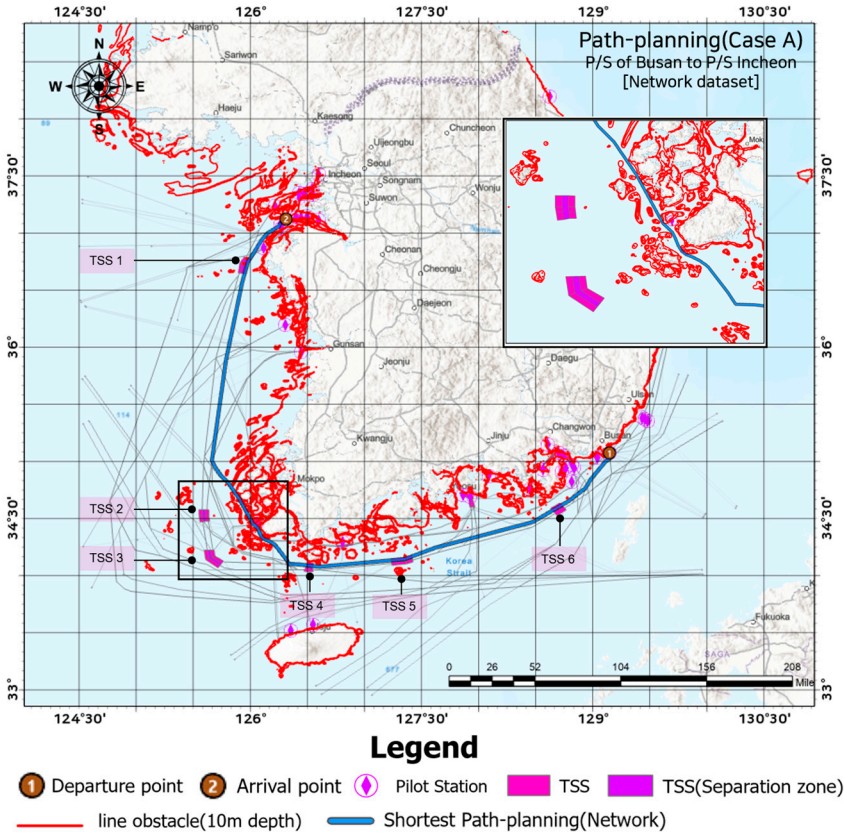

**Figure 12.** Path planning of the network dataset (Case A) from Busan to Incheon.

Table 3 presents a comparison of the path planning of the HANBADA ship and the shortest distance path generated by the network dataset in terms of distance, sailing time, and compliance with TSS regulations. The distance of HANBADA's path planning (Busan to Incheon) is 708.5 km, and the sailing time is analyzed to be 00 d 21 h 50 m when sailing at the service speed of 17.5 knots. Furthermore, the current ship operation plan is determined based on the captain's decision, and the ship may sail on routes other than the shortest distance. The analysis showed that the HANBADA ship passed through all six regulated TSSs in Korean coastal waters. Furthermore, the shortest distance path generated by the network dataset is analyzed to be 663.9 km, and it takes 00 d 20 h 28 m of time when sailing at 17.5 knots, which is the same speed as HANBADA. The shortest path of the network dataset navigated between the islands without passing through TSS 2 and TSS 3, but complied with all other TSS regulations and provided path planning. Therefore, the shortest distance path planning of the network dataset reduced the distance by 44.6 km relative to the HANBADA path planning, enabling arrival 01 h 22 m sooner.

**Table 3.** Comparison of path planning (Case A) between HANBADA and network dataset.

| Category (Case A) | Distance to Go (km) | Sailing Time (Distance/17.5 knot) | Comply with TSS 1 | Comply with TSS 2 | Comply with TSS 3 | Comply with TSS 4 | Comply with TSS 5 | Comply with TSS 6 |
|---|---|---|---|---|---|---|---|---|
| HANBADA | 708.5 km | 00 d 21 h 50 m | Comply | Comply | Comply | Comply | Comply | Comply |
| Network dataset | 663.9 km | 00 d 20 h 28 m | Comply | No | No | Comply | Comply | Comply |
| Difference | −44.6 km | −00 d 01 h 22 m | - | - | - | - | - | - |

Figure 13 shows the path planning of the training ship, whereas Figure 14 shows the path planning based on the network dataset, connecting the P/S of Incheon and Busan, and returning to the starting point.

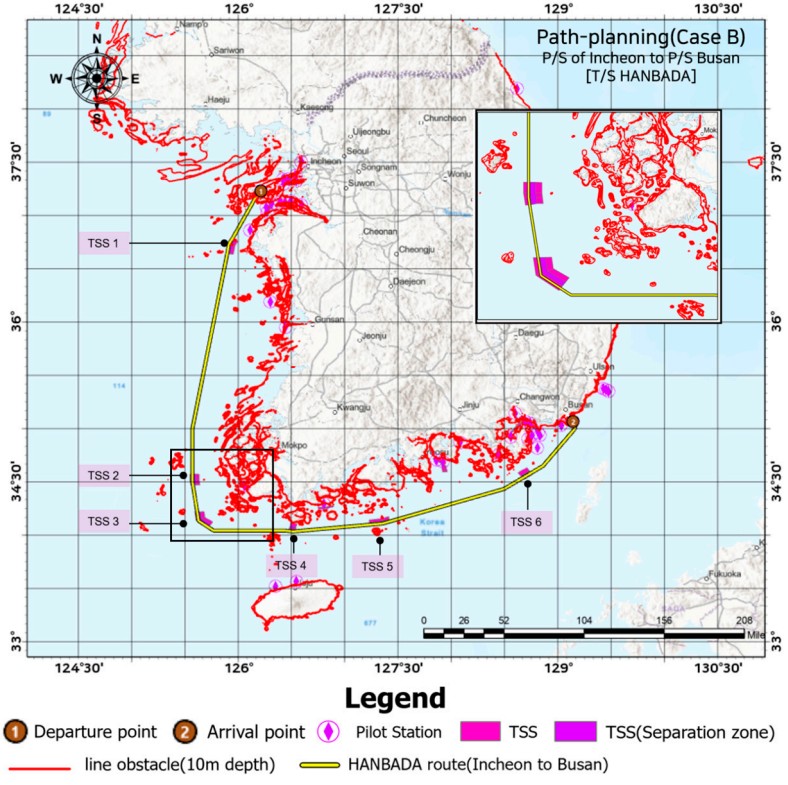

**Figure 13.** Path planning of the training ship (Case B) from Incheon to Busan.

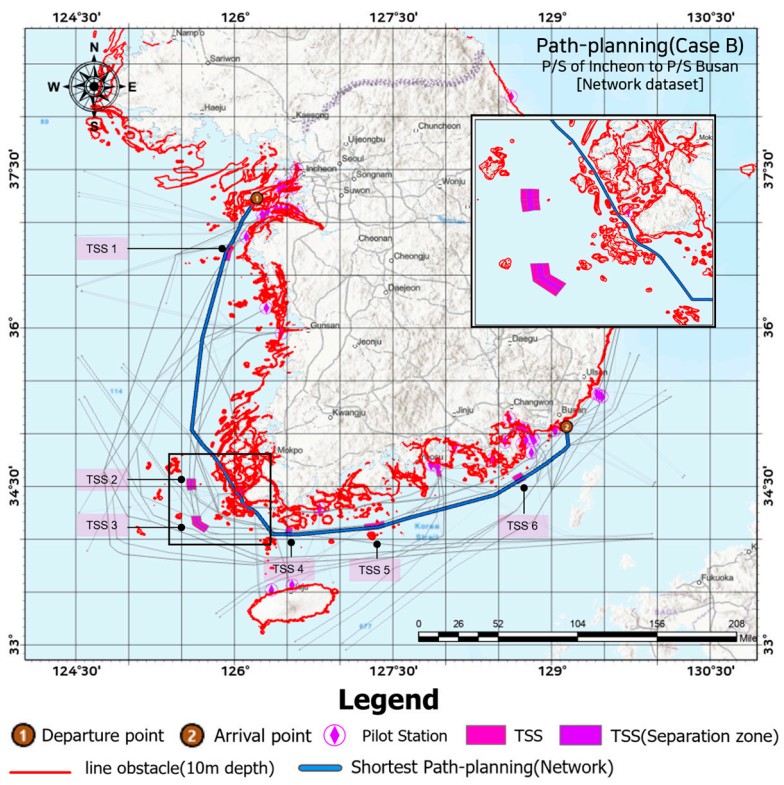

**Figure 14.** Path planning of the network dataset (Case B) from Incheon to Busan.

Table 4 compares the distance, sailing time, and compliance with TSS regulations of the HANBADA path planning and the shortest route provided by the network dataset for the return trip from Incheon P/S to Busan P/S. The HANBADA path planning covers a distance of 723.7 km and has a sailing time of 00 d 22 h 21 m when sailing at 17.5 knots, and complies with all regulations from TSS 1 to TSS 6. In contrast, the shortest route provided by the network dataset covers a distance of 680.7 km and has a sailing time of 00 d 21 h 02 m. Because the shortest route sails between the islands, it does not pass through TSS 2 and TSS 3. In summary, the results show that the network dataset can reduce the distance by 43 km and the sailing time by 01 h 19 m, while HANBADA's path planning complies with all TSS regulations.

**Table 4.** Comparison of path planning (Case B) between HANBADA and network dataset.

| Category (Case A) | Distance to Go (km) | Sailing Time (Distance/17.5 knot) | Comply with TSS 1 | Comply with TSS 2 | Comply with TSS 3 | Comply with TSS 4 | Comply with TSS 5 | Comply with TSS 6 |
|---|---|---|---|---|---|---|---|---|
| HANBADA | 723.7 km | 00 d 22 h 21 m | Comply | Comply | Comply | Comply | Comply | Comply |
| Network dataset | 680.7 km | 00 d 21 h 02 m | Comply | No | No | Comply | Comply | Comply |
| Difference | −43.0 km | −00 d 01 h 19 m | - | - | - | - | - | - |

## 5. Discussion

The increase in maritime traffic and vessel size has significantly changed the marine spatial planning (MSP) environment [40]. As the demand for various marine activities increases, the area covered by maritime traffic routes decreases. Simultaneously, the importance of maritime traffic networks for the smooth transportation and handling of cargo is increasing. The shrinking area of maritime routes and the increasing importance of networks signify the complexity of the marine environment. Furthermore, the future

development of MASSs and USVs anticipates the operation of autonomous maritime transportation without human decision making. Fundamentally, compliance with predetermined routes is a priority for an unmanned MASS to operate from the point of origin to the destination. In this case, the predetermined routes are safe and can be used to determine the shortest distance. Therefore, analyzing the safe conditions and determining the shortest distance are necessary. The safe and shortest route mentioned here can be determined by analyzing the operating patterns of ships equipped with an AIS. If AIS data accumulated over several years can be used to identify areas where ships frequently operate, this route implies safe and economical results [41,42]. Therefore, maritime traffic routes, maritime traffic networks, and shortest path planning should be based on AIS data. In this study, we differentiated the following characteristics to construct a maritime traffic network dataset for analyzing shortest path planning.

Areas where ships frequently navigate and where AIS data have been accumulated over several years can be selected as safe and economically efficient routes. Accordingly, a maritime spatial-temporal density analysis was performed to extract the areas where ships operate the most.

A line that can induce ship flows is required to separate the flow of maritime traffic within a density-based polygon. The IMO regulates TSSs for densely trafficked areas, which separate traffic flows in a two-way direction. The Delaunay triangulation algorithm is used to achieve this within the polygon. Two-way traffic flow can facilitate preventing collision accidents because ships operate in a consistent direction, and this method can be effectively applied to future navigation plans for MASSs and USVs.

To construct the network dataset, the safety depth values based on a digital chart (S-57) were included, and the shortest path planning algorithm based on Dijkstra's algorithm was proposed. The results of shortest path planning were compared with the actual route used by the training ship. One distinctive feature of this study is that it compares the actual route used for navigation rather than comparing various algorithms for finding the shortest path. A limitation of this study is that, while considering factors such as water depth and designated routes utilized by ships, it did not account for various environmental factors in the sea, such as weather conditions, sea states, and currents. Additionally, the study did not encompass various aspects of maritime conditions, including navigation rules, encounters between vessels, and collision avoidance, which are crucial for ship operations.

## 6. Conclusions

This study contributes to the literature by providing an understanding of shortest path planning based on existing spatial-temporal density analysis results and network datasets. The main significance of the research findings is that the maritime traffic network dataset enables ships to be used in an easy, time-efficient, and safe manner for determining the shortest path. The comparison between the maritime network dataset and the shortest path planning for the training ship revealed the superiority of the AIS data-based network dataset. The comparisons for both Case 1 and Case 2 indicated a reduction in sailing time by over one hour and a distance decrease of more than 40 km. Maritime traffic networks are important; therefore, various countries are actively conducting analyses using different methods. Based on previous studies, this research proposes a novel approach to extract polygons using spatial-temporal density, separates traffic flows in a two-way direction, and suggests safe and economical shortest path planning using digital chart depth values. However, this study has a limitation in that it requires an accurate knowledge of maritime laws and navigation patterns. Although the traffic flow was separated in a two-way direction, the actual traffic environment at sea is considerably more complex and diverse. Developing a maritime traffic network requires combining various elements, such as ship encounters, safe speed, TSS, and overtaking. However, the current network dataset only considers TSS and safety depth, which is a limitation. Future research should aim to develop a maritime network dataset by combining these factors. Furthermore, there is

a need to advance research by considering and incorporating a broader range of marine environmental factors, including weather conditions, sea states, and currents.

The analyzed area focused on Korea's coastal waters, which generate various trade ports and hundreds of routes; however, this area has the potential to be connected globally. Accumulated AIS data over several years are a valuable resource for inferring safe and economical routes. If a maritime traffic network dataset is constructed using the AIS data from all maritime areas, it will be useful to ship operators, cargo managers, shipowners, and government officials. This topic is of great interest when considering the potential use of MASSs or USVs as unmanned vessels in the future.

**Author Contributions:** Conceptualization, J.-S.L. and T.-H.K.; methodology, J.-S.L.; software, J.-S.L.; validation, Y.-G.P., T.-H.K., and J.-S.L.; formal analysis, J.-S.L.; investigation, J.-S.L.; resources, J.-S.L.; data curation, T.-H.K.; writing—original draft preparation, J.-S.L.; writing—review and editing, Y.-G.P.; visualization, T.-H.K.; supervision, Y.-G.P.; project administration, J.-S.L.; funding acquisition, T.-H.K. All authors have read and agreed to the published version of the manuscript.

**Funding:** This research was supported by Korea Institute of Marine Science & Technology Promotion(KIMST) funded by the Ministry of Oceans and Fisheries(20200495, "Development of satellite based system on monitoring and predicting ship distribution in the contiguous zone.

**Institutional Review Board Statement:** Not applicable.

**Informed Consent Statement:** Not applicable.

**Data Availability Statement:** This research was conducted with data provided by the Ministry of Ocean and Fisheries.

**Conflicts of Interest:** The authors declare no conflict of interest.

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
