# Peer review of "Maritime Transport Network in Korea: Spatial-Temporal Density and Path Planning"

_jmse, doi:10.3390/jmse11122364_

Round 1
Reviewer 1 Report
Comments and Suggestions for Authors
The proposed method in this paper for constructing a maritime route network dataset based on spatial-temporal density analysis and Delaunay triangulation has merit. However, there remain some issues that could be addressed through the following recommendations:
1. Construct and validate the networks across multiple regions using AIS data encompassing more months.
2. Analyze the computational performance under network datasets of varying complexity. Consider leveraging parallel algorithms to enhance efficiency.
3. Comprehensively consider more factors impacting navigation safety, such as visibility and current velocity. Construct a multi-source heterogeneous information system to provide safety alerts.
Reviewer 2 Report
Comments and Suggestions for Authors
The original achievement of the authors of the paper is the presentation of a method for planning the shortest ship route based on the routes of other ships previously recorded by the AIS system, taking into account the density of their traffic.
Comments:
1. The presented study is more of a research report than a scientific article.
2. The Introduction lacks the formulation of the thesis and the resulting objectives of the work, which would become its chapters.
3. The influence of hydrometeorological conditions on the ship's route was completely ignored.
4. Previously recorded ship routes in various environmental conditions cannot be a reference to current navigation conditions.
5. Complete the paper with an illustration of the developed ship route planning algorithm in the form of pseudocode and a description of the software used.
6. In the Conclusions, there are no specific quantitative and qualitative conclusions from the research conducted.
7. The content of the last paragraph from Discussion should be moved to Conclusions.
8. Supplement the Conclusions with a detailed plan for further research on the topic of the paper.
